# Evaluation of Railway Station Infrastructure to Facilitate Bike–Train Intermodality

Margherita Pazzini [1],*, Claudio Lantieri [1], Annalisa Zoli [1], Andrea Simone [1] and Hocine Imine [2]

1 Department of Civil, Chemical, Environmental, and Materials Engineering (DICAM), University of Bologna, 40131 Bologna, Italy
2 Laboratoire sur la Perception, les Interactions, les Comportements et la Simulation des Usagers de la Route et de la Rue (PICS-L), Components and Systems Department (cosys), Gustave Eiffel University, 77420 Champs sur Marne, France
* Correspondence: margherita.pazzini2@unibo.it

**Abstract:** In recent years, emissions into the atmosphere have been brought to the attention of the authorities and some action has been taken to try to solve the problem. One is the application of EU legislation 2008/50/EC, which requires states adhering to this law to constantly monitor air quality and subsequently find solutions to reduce the impact of emissions. The data show that 20% of emissions come from transport, 70% of which come from private vehicles. Sustainable mobility can be a possible solution to reduce pollution and traffic congestion. The promotion of cycling, as part of sustainable mobility, is a required action to achieve the objectives pursued. This research aims to define the quality of infrastructure and accessibility of railway stations to the use of bicycles. The approach used was to define a technical checklist to estimate the criticalities of the structure in a quantitative way. An example is the case study developed in the Emilia-Romagna region (Italy) within the PREPAIR project where 33 railway stations were classified and analyzed. In the end, the checklist was effective in defining the necessary interventions and the required activities and can be used in similar cases during the decision-making processes.

**Keywords:** intermodality; accessibility; cycle paths; railway stations; sustainable mobility

## 1. Introduction

Reducing $CO_2$ emissions into the air is a global issue that major authorities have been considering for years. As to this purpose, the International Energy Agency has committed 30 member countries to achieve zero emissions by 2050 [1]. Moreover, it should be stressed that an important part of emissions comes from the transport sector. In fact, the 2019 figures show that $CO_2$ emissions from transport account for 25.2% of total emissions, 17.5% of which come from road transport [2], and significantly, about 70% from passenger cars [3]. In Italy, daily travelers amount to 30,214,401, 50.7% of the total number of citizens. In particular, 67.9% travel for work and 32.1% for educational reasons. A total of 57.5% of these people travel within their municipality, while 42.5% move outside of it [4]. The data concerning commuters are extremely important as they indicate the large number of people who rely on the mobility infrastructure every day [5]. A survey from the University of Trieste shows that 70.1% of all commuters use their private car to reach their destination (17.5% use only private cars, 52.6% use their car combined with a public transport) [6,7]. To improve this situation, a possible solution could be to replace 20% of private cars, thus reducing air emissions by 12% ($CO_2$ equivalent). Of course, to achieve this purpose and significantly reduce pollution, private car users should rely on an efficient network of multiple and shared means of transport, including bicycles [7–9] with mobility hubs allowing for easy and safe access to different modes of transport [10–12]. Relying on an intermodal transport means that the destination is reached using at least two different modes of transport, while providing economical and technical advantages, improving

efficiency and reducing energy consumption and greenhouse gas emissions [13–15]. To this extent, not only do mobility hubs help in decreasing emissions, but they also provide a healthier lifestyle by reducing accidents and facilitating connections [15]. Moreover, it was demonstrated that favoring intermodality improves the attractiveness and efficiency of a journey [16]. Mobility hubs and real global hubs should be located in strategic areas, such as railway stations; the infrastructures should be grouped together, thus allowing for users to reach public and shared transport as well as the city centers [9,17,18].

Non-motorized transport modes, such as bikes and kick scooters, which actively involve pedestrians and cyclists, are among the most important means of transport to be found at mobility hubs. In fact, they are cheap, noise-free, environmentally friendly and can be parked in small places [19,20]. It should be noted that the European Decree 2008/50/EC itself defines bicycles as a sustainable means of transport [21]. Moreover, the Sustainable Urban Mobility Plan (SUMP), a European strategic plan based on the infrastructure planning of the mobility of goods and people to increase the quality of life in and around cities, also defines bicycles as non-polluting means of transport. The grouping of bicycles, public transport and walks results in the description of perfect sustainable mobility, but it only counts for less than 40% of all daily trips [18]. The opportunities created by the development of bicycle accessibility to reach railway stations and the facilities provided are therefore of great importance [22].

Encouraging travelers to combine the use of bicycles with the use of trains is a challenge that involves also the infrastructure around the railway station. It has been proven that a better quality of cycling infrastructure has a positive impact on the utilization of this means of transport, thus increasing the number of users [23]. As for the use of bicycles, in addition to the quality of dedicated infrastructure, speed limits of less than 30 km/h, traffic-calming devices, the safe use of common ground, proximity to suitable trails, and adequate slopes [24,25] should be guaranteed to further attract cyclists [26]. In addition, the volume of traffic, the risk of accidents, the distance to the central business district, the high-density housing areas, the type and quantity of cycling facilities, road connectivity, population density and the perception of infrastructure availability are important factors to consider [27–30]. Maria Konstantinidou et al. [31], through a preference questionnaire survey, found that the propensity to cycle depends on the presence of cycle paths, parking areas, bike-sharing facilities and on the quality of air breathed while cycling. Moreover, Maciorowski et al. [32,33] consider bicycle paths, crossing options, lighting and signaling, afforestation and shading as part of the infrastructure that users of non-motorized vehicles consider while travelling. Krizek and Stonebraker [33], through a survey using the Analytic Hierarchy Process (AHP) multicriteria decision-making tool, found that cyclists attach great importance to bicycle parking security, while Egan et al. [34] found that bicycle parking types defined as "Any", "Accessible" and "Secure" are preferred, placing emphasis on locked or guarded bicycle parking facilities. Finally, Arbis et al. [35] found that the favorite spot to park a bicycle depends on the frequency of the railway service. When the train service is poor, the chances of theft are greater and bicycle users prefer to park their bicycles into locked spots. Given the vulnerability of accidents, cyclists' safety is one of the most important parameters to consider when analyzing the use of this means of transport [36–38]. Many studies have analyzed the number of accidents related to infrastructure or to mixed traffic of cars and non-motorized vehicles. For example, Osama and Sayed noted that the number of cyclist–motorist crashes has no linear relationship with the increase in vehicles and transit traffic or with socio-economic variables, as well as variables related to the built environment. However, it has been shown that the decrease in cyclist–motorist crashes is related with the increase in the proportion of local roads and off-street bike links, as well as an increase in recreational and residential density [39]. Besides cyclist–motorist crashes, there are also single-bicycle crashes (SBC), which count for 85% of all crashes involving bicycles. By diving deep into the causes of SBC, it was found that 44% is due to road maintenance deficiencies, 16% to bicycle–cyclist interaction, 15% to road design, 14% to cyclist behavior and 10% is due to the interaction with other road users (e.g., evasive

actions to avoid collision) [40]. The above data again indicate the importance on acting on the quality of the infrastructure.

Different methods define in the literature the quality of the cycling infrastructure. Tran et al. defined the accessibility and performance metrics of mobility hubs based on different weightings to reflect different policy priorities. The output is infrastructure investment, servicing and maintenance.

As for Italy, actions have been taken to reduce emissions through the application of the Legislative Decree of 16 June 2022, n. 68, which provides founds to improve road safety, including cycle paths [41]. The "PREPAIR" project—Po Regions Engaged to Policies of AIR—was founded by the Life European program in 2016 with the aim of implementing the measures provided by the Air Quality Plans and to achieve the "Bacino Padano" (Po Basin) Agreements on a larger scale [22]. The objective of this article is to define a methodology to give priority to infrastructure needs and interventions to facilitate the accessibility of bicycles in railway stations. Cyclist parameters are considered to be important both in the literature and in the Italian decree of 11 January 2018 n.2, and have been taken into consideration. A case study applied in the Emilia-Romagna region, Italy, for the accessibility of bicycles inside 33 railway stations, was carried on in order to develop intermodality between bicycles and trains. The methodology will provide a ranking of results to demonstrate the most performing and, above all, the least efficient railway station. It will also show in which area train stations are less effective and how to implement it. Figure 1 shows the different steps followed to obtain the monitoring table definition and the main critical issues of the bike–train intermodality infrastructure.

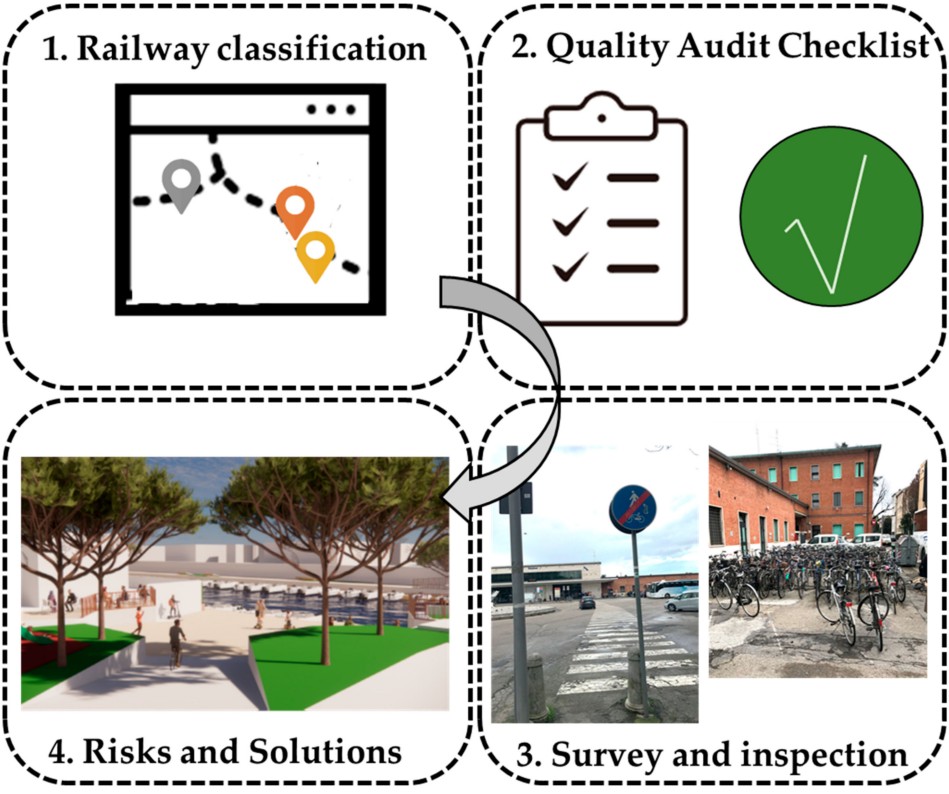

**Figure 1.** Steps to obtain the monitoring table definition to determine the critical issues in the bike–train intermodality.

## 2. Case Study and Railways Classification

This project is driven by the necessity to define the quality and interventions necessary for the accessibility of bicycles inside mobility hubs in order to increase intermodality between bicycles and trains. This aim is part of PREPAIR, which is a project involving Italy and Slovenia based on achieving the objectives of the Air Directive by implementing the Air

Quality Plans (Directive 2008/50/EC) [21]. One of the main objectives of this plan concerns "actions to promote cycling mobility" with the intention of reducing vehicle congestion and air pollution in urban areas. This part includes the section "survey on bike infrastructures availability in railway stations", stating the importance to have good bike infrastructures in the main railway stations of the study area and to understand the amount of investment that should be made to reach satisfying levels of efficiency. To achieve this goal, bicycle–train intermodality needs to be analyzed to detect the actual deficiencies and eliminate them. The measures proposed by PREPAIR are taken by municipalities with more than 30,000 people, that is, by those who are obliged to adopt the Urban Traffic Plans. In Italy, this task is led by the Emilia-Romagna region, where the applied case study is based.

*2.1. Case Study*

The Emilia-Romagna region is located in the northern part of Italy with a population of 4,459,866 in 2021 [42]. The region borders Lombardy, Veneto, Piedmont, Liguria, Tuscany, Marche and the Republic of San Marino, while the eastern side overlooks the Adriatic Sea. Due to its geographical and historical features, the region is a tourist attraction and offers a multitude of job opportunities. In 2021, the Emilia-Romagna region hosted more than 30 million tourists, some of them travelling within the cities, attracted by business trips or spa facilities [43]. The presence of the Apennines and of the hills attracts tourists for outdoor activities as well. Moreover, the Emilia-Romagna region, characterized by the coast of the Adriatic Sea, reaches the highest share of tourists in this area during the summer thanks to equipped beaches and activities involving different age groups. These data show the importance of movements between different localities inside the region, which also depend on the season of the year. The Po Valley is also part of the Emilia-Romagna region. Crossed by the river Po, the main Italian river, this is historically one of the most important industrial areas of northern Italy. This leads to a substantial movement of commuters who need to move exclusively for work reasons to areas far from their households. The railway station infrastructure, as a mobility hub, is therefore of great importance and must include all the facilities necessary to reach the place of interest in the easiest and most sustainable way possible, without excluding bicycles. To give priority to interventions, the following method has been developed.

*2.2. Classification of Railway Stations*

The project starts with the classification of the railway stations by RFI—The Italian Railway Network—which divides each railway station into three categories: gold, silver and bronze. These categories are important to define which features and services should be included in a railway station in relation to the number of passengers attending it, the level of services offered to the travelers, the presence of areas opened to non-travelers and the intermodality present in the node. Figure 2 shows the categories defined.

- GOLD: These railway stations have an average of 10,000 visitors per day and offer services to travel for long and short distances. Inside the railway station, there are also services and facilities for non-travelers and sometimes for the city. The number of gold railway stations present in the study area is 10.
- SILVER: These railway stations have an average daily attendance ranging from 2500 to 4000 people and may offer services to travel for long and short distances or only regional and metropolitan services. The number of silver railway stations present in the study area is 8.
- BRONZE: These are the smallest railway stations. They have 500 visitors per day and are only provided with regional and urban services. The number of bronze railway stations present in the study area is 15.

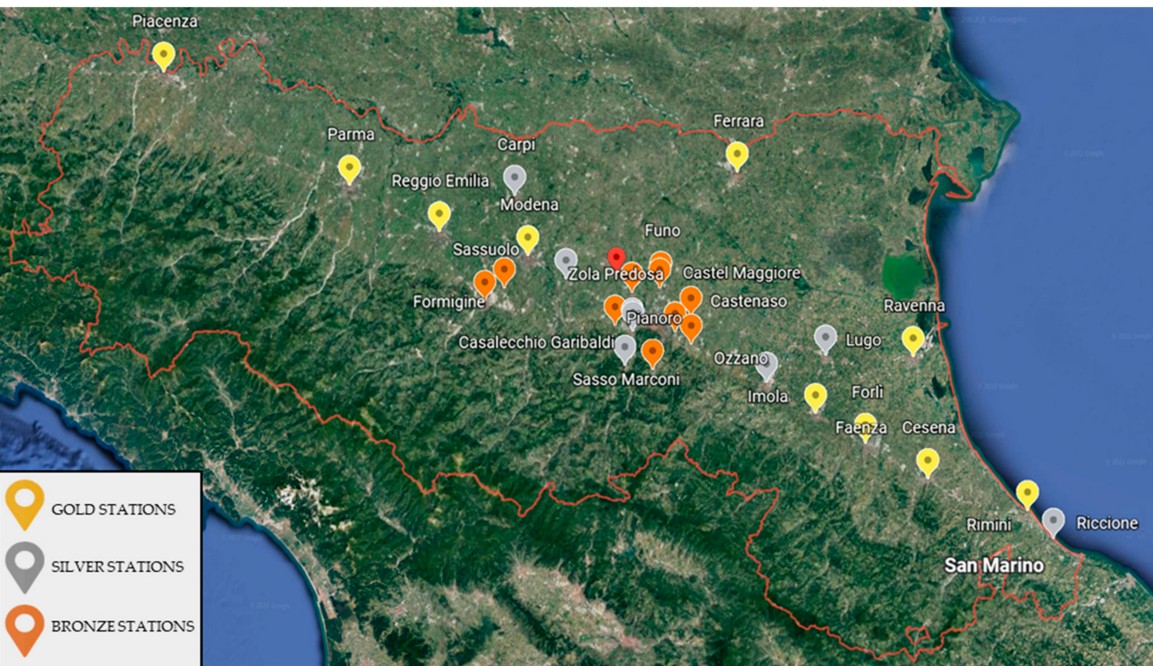

**Figure 2.** The 33 railway stations of the Emilia-Romagna region analyzed (© Google Earth).

The parameters used to create this classification were:

- Daily attendance: this is given by the number of passengers who daily pass through the station to get on or off a train and by the number of people who, although not using the passenger transport service, still frequent the facility (for purchases, interchange with other types of transport, tourism, simple transit, etc.);
- Level of passenger service: this considers the importance of the system exclusively in terms of the commercial offer of transport, counting the number and type of trains (AV-high-speed train, long-/medium-distance train, regional or metropolitan train);
- Areas open to the public: this criterion considers the total area of services open to the public; commercial areas such as shops and exhibition areas; transit areas; underpasses; transit tunnels; so-called "operational" areas, that is, those areas that from outside the station lead passengers to the train (platforms, main entrances, track header, ticket offices);
- Intermodality: this considers the simultaneous presence or absence—within the station or in the immediate vicinity—of metro stops; bus terminals for urban/extra-urban buses; tram stops; taxi lanes; connections to airports; car, motorcycle and bike parking lots.

Figure 2 shows the 33 mobility hubs of the 30 municipalities of the Emilia-Romagna region that took part in PREPAIR.

## 3. Methods

A checklist was created to understand the facilities at each railway station. The checklist was integrated with the elements included in the regional guidelines and in The Highway Code. The use of the checklist allows for each railway station to be compared with the most performing, ideal one—namely, the one that obtains the highest score in all the sections of the checklist—as well as to compare stations among them. The inspection in the checklist begins by analyzing the cycle path that ends near the railway station. If there are no bike paths, the speed limit of 30 km/h is checked. In fact, a speed limit of 30 km/h or less indicates that bicycles and cars can safely proceed together on the same road. In order to assess the safety of a cycle path, the presence of both vertical and horizontal signs indicating a possible cycle path should be analyzed. In addition, the visibility of such signs must be assessed as well. Another important feature to analyze is the type of cycle path, if promiscuous or separated from pedestrians. The inspection then assesses the services

provided to cyclists, such as a safe storage for bicycles or a bike-sharing service. Each of these attributes is divided into detailed sections to dive deep into the characteristics of the accessibility of bicycles in the proximity of the railway station and inside it.

Appendix A contains the checklist used to carry out the analysis on the accessibility of cycling infrastructure within railway stations that considered the D.M. of the 30 November 1999, n.557 (regulation with the rules for defining the technical characteristics to build cycle tracks) [44], the "New highway code" [45] and the regional guidelines [46].

The monitoring tables are divided into five main groups that are: the quality of the cycle path infrastructure regarding access to the railway station, the sign and lighting, the services provided to the cyclist, the bicycle safekeeping service and the bike-sharing service.

### 3.1. Cycle Path Infrastructure Regarding Access to the Railway Station

The section regarding "cycle path infrastructure in access to the railway station" includes the following points:

- The presence of a cycle path and its distance from the railway station. These characteristics refer to Art.2 of the Regional Law of the 5 June 2017, n.10, which promotes cycling through interventions and actions aimed at improving the quality of daily trips [47]. The cycle path close to the railway station is a great way to promote active mobility.
- Assessment of the type and quality of pavement that together ensure the safety of users. In particular, the adherence of the pavement and the presence of plant roots, or anything else consuming the pavement should be detected. Rate "2" is given to an "excellent" pavement, "1" to a "good" pavement and "0" to a "bad" one.
- Attachment 3 of the Regional Guidelines states that, in urban areas, two-way cycle paths are not allowed since safety would not be guaranteed. This is the reason why rate "0" is assigned to two-way cycle paths and "1" is given to the one-way kind. Moreover, rate "2" is given to paths reserved for bicycles only, "1" to streets with a speed limit 30 km/h and "0" is given in case the type of cycle path is unclear.
- Geometrical standards defined by the law and guidelines [48]. They include the width of the path, which should be 1.5 m only in the case of a one-way path; the minimum width of 1.25 m for each side of the path just in case of more than one cycle path; if the cycle path width is reduced to 1 m, it has to be indicated.
- The slope of the cycle path. This is an important factor to engage bike users and should be less than 5% to limit speed and to enable riding uphill for less agile users.
- One-off obstacle warning. They can reduce the width of the bike path and careful attention should be paid not to reduce safety for users.
- The presence of pedestrians, vehicles or lateral entrances. In particular, conflicts between bicycles and pedestrians are evaluated, as well as possible users from side streets, shops or garage doors that could interfere with the safety of the cyclist.
- Visibility of the access to the cycle path on exit from the railway station. If the paths are not visible, cyclists could reduce their safety by using another option, such as a different road or a different means of transport.

### 3.2. Signs and Lighting

The "signs and lighting" section is defined by the following technical legislation: "Technical instructions for the design of cycle networks—Draft n.3 of the 17 April 2014", the "Paper of the interministerial decree of the 31 March 1993—n.432" and the "New highway code". The signs are also divided into horizontal and vertical signs, and the elements on which the method focuses on are:

- Horizontal signs on the cycle path. These must be clearly distinguishable from other road signs using a visible color. The brightness of the signals is evaluated by assigning the number "2" in case of good color, "1" to a medium color and "0" to no color. The edges of the cycle path must also be visible and ensure safety for users.

- Vertical signs near the railway node and along the cycle path. Such signs should be present to provide users with important information. The New Highway Code requires the presence of "the cycle lane sign contiguous to the sidewalk". It must be installed at the beginning (and at the end) of the cycle path. A sign indicating the pedestrian and cycle path should be set at the beginning (and end) of a promiscuous cycle path. Signs within the railway station to direct cyclists inside are also evaluated.
- Lighting is analyzed and its source is checked. Rate "2" is given in case the lighting is dedicated only to the cycle path, rate "1" in case the lighting is on the general road, and rate "0" in case it comes from advertising or no lightning is provided.
- Signs indicating the points of interest of the city must provide users with all the "necessary information for a proper and safe circulation, as well as easy identification of routes, locations and services" (Art. 124 C.1 RA) (Art. 124 (Art. 39 Cod. Str.). Rate "2" is assigned if the signs are present and complete, "1" in case they are incomplete and "0" in case there are none.

### 3.3. Services Provided to the Cyclist

The third section includes the "services provided to the cyclist". The points in this part mainly refer to the "Regional guidelines for cyclability" [46]. It is extremely important that a railway node is furnished with bicycle racks, located at a short distance from the station. Besides securing a bicycle wheel, they should also fix the bicycle frame. The parameters used to define this section are:

- The bike racks covered by shelters are evaluated based on the number of bicycles they can hold.
- The bike racks not covered with shelters are evaluated depending on the portion of bicycle they can secure. If they can secure the body and the wheel, they are marked with number "1"; if they can only secure the wheel of the bicycle, they are marked with number "0".
- Accessibility: the presence of elevators or ramps is evaluated, allowing users to carry a bicycle and reach the platforms comfortably, or on the contrary, they can only use stairs.
- Toilets inside the railway station are important to guarantee comfort. Toilets inside commercial activities have not been considered.

### 3.4. Bicycles Safekeeping Service

The safekeeping places for bicycles are large areas closed to the public that offer the possibility to leave a bicycle safe from all weather conditions and theft, since they are often guarded by personnel. Moreover, most of the time, they are paid services and are accessible using a key or magnetic card.

The following parameters are especially evaluated:

- Proximity to the train station. In this case only, the presence of a safekeeping service for bicycles is considered useful, otherwise it loses attraction to the user.
- The availability of the spots for bicycles inside the safekeeping area is also rated based on the needs.
- Any additional services, such as toilets and vending machines, provided to the user within the safekeeping place.

### 3.5. Bike-Sharing Service

The section regarding bike sharing is evaluated considering its presence, the distance from the railway node and the availability compared to the needs. Moreover, the possibility to leave the bicycle in a spot different from the point of collection is evaluated with the number "1", while number "0" is given to its opposite.

### 3.6. Checklist Score Calculation

At this stage, the checklist created only provides a qualitative result. In order to transform it into a quantitative evaluation, the road network safety ratings [49] and the

checklist proposed by the Agency for the Control and Quality of Public Services in the area of Rome [50] were examined. In the study from Rome, the checklist is implemented by assigning a number to each characteristic based on the criticalities that the on-site inspection identified, contributing to the creation of a monitoring table; here, the inspection focuses on the quality of cycle paths in Rome. The road network safety rating is used to monitor the safety quality of a road network and to act on the inadequate ones. Both methods use a checklist where a number is assigned to each topic. In the checklist elaborated for the inspection of intermodality between bicycles and trains, the numerical evaluation of each section is given using an algorithm that provides the following results:

- Every main section is rated between 0 and 100;
- The positive answers provide 1 point, the negative ones provide 0 points;
- The multiple-answer responses provide a fractional rate between 0 and 3;
- The values obtained are, at the end, given in a percentage by multiplying the total number of questions in the single section by the values obtained in the single question in relation to the maximum achievable value.

The calculation is carried on:

$$\% \ section \ score = \frac{100}{n_{tot}} \left( \frac{x_1}{x_{1tot}} + \cdots + \frac{x_j}{x_{jtot}} \right) \tag{1}$$

- $n_{tot}$ = the number of questions in the single section;
- $x_1$ = the score of the first question;
- $x_{1tot}$ = the maximum score of the first question;
- $x_j$ = the score of $j$ question;
- $x_{jtot}$ = the maximum score of $j$ question.

- The monitoring tables, one for each type of railway station, are divided into five main groups that consider the above-mentioned characteristics.

The checklist is completed during the inspection while the results are calculated in a second phase, providing a result for each section of each station in the form of a percentage.

## 4. Results

The results obtained for each section of the checklists are represented in Figure 3 by comparing the three different station categories.

"Sec.IV", concerning the quality of the bicycle safekeeping service, has the worst results in all three station categories. This service is not present in any of the bronze stations and exists only in a single silver station; even in the gold stations, it does not achieve satisfactory results, as it is estimated at only about 40%.

"Sec.V", concerning bike sharing, is present in only one bronze station, while it is present in two silver railway stations where the quality of the service is valued at 75%.

When comparing the quality of each gold station, it turns out that bicycle safekeeping services are those that need the most improvements, both inside and around the train station. In fact, this service is missing in three of the railway stations defined as gold and, where present, it does not reach a quality percentage above 86%. The presence of a greater amount of bike-sharing reduces the problem related to bicycle safekeeping. Signs and lighting ("Sec.II") in proximity to the railway station node represent the second category to be improved in the gold railway stations. Here, only one railway station scores more than 80%, while 4 railway stations out of 11 score less than 50% of quality. The remaining three sections, "Sec.I", "Sec.III" and "Sec.V" for gold stations are rated between 70% and 80%. "Sec.III" must be stressed as having two railway stations that reach only 50%.

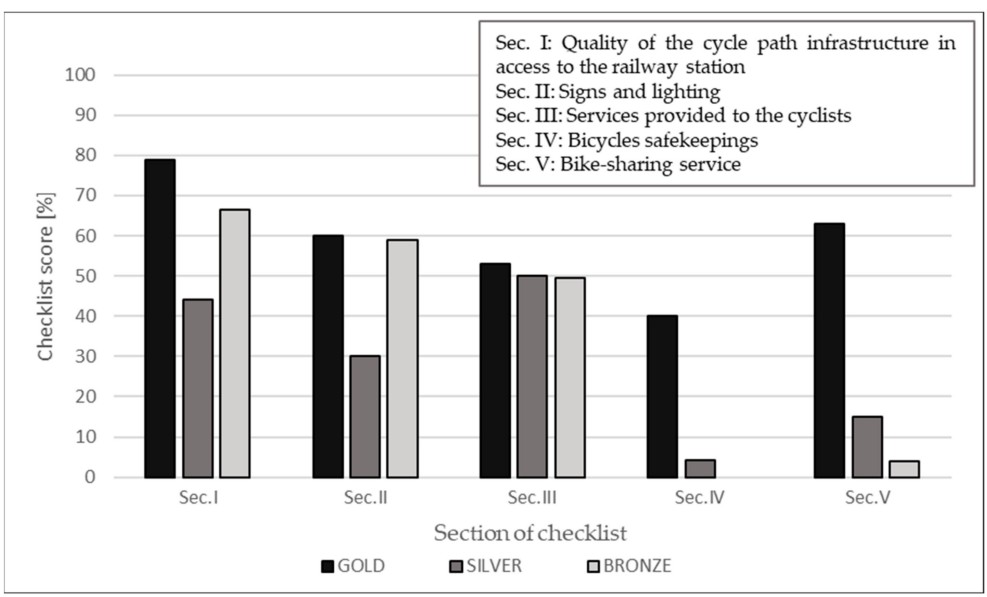

**Figure 3.** The average results for every section of each station category.

Compared to the other categories, the silver and bronze railway stations are mostly lacking in bicycle safekeeping services ("Sec.IV"). The mostly unfinished category, and therefore, to be implemented for bronze and silver railway stations, is "Sec.V", which shows the rare presence and low quality of the bike-sharing service. The most performing section in silver railway stations includes services provided to the cyclists. However, they reach only 70%. "Sec.I" and "Sec.II" are rated below 50% in silver train stations. On the contrary, "Sec.I" and "Sec.III" are the most performing sections in bronze railway stations, with a small difference between the two of 66% and 63%, respectively. Finally, Figure 4 shows that the score of "Sec.III" does not differ much among the three gold, silver and bronze categories, respectively, 71%, 66% and 63%.

In the next phase of the study, the results obtained for the three categories of stations in each of the check list applications were compared. Figure 4 shows the results obtained in percentage terms in the five sections analyzed. For the three categories of stations, the trend obtained in each graph is similar because the highest percentages and most critical issues are in the same points.

In Sec.I, concerning the quality of the cycle path, gold stations achieved the highest results in most of the questionnaire items, obtaining 100% in four questions. Question 5, regarding the type of cycle path (Appendix A), obtained the lowest value for all the three categories of stations because the cycle path—if present—is of bidirectional type. Another problem faced for all three types of stations is the conflict with pedestrian traffic (question 12) due to the presence of promiscuous cycle paths. The quality of cycle path is generally worse for silver stations, which never obtain average values above 50%. Basically, this is because only four out of eight stations belonging to this category have a cycle path. The score obtained by each single silver station with a cycling infrastructure ranged from 80% to 90%.

In Sec.II, concerning the visibility and signage of the cycle path, the low values obtained are due to the coloring of the bike path pavement, especially in gold and silver stations. In fact, where a cycle path is present, it is not visible near the railway stations because it is poorly marked. Even vertical driving signs for cyclists are inadequate at most stations. In fact, the signage reaches only 40% for gold stations, while for bronze and silver stations just 20%. On the contrary, gold stations have good night lighting, although this does not always come only from the bike path but also from street lighting.

Sec.III, relating to services for cyclists near the railway node, shows similar values for all three categories of stations. In fact, parking spaces for bicycles outside the station are inadequate for all the stations analyzed. In the bronze stations only, there are shelters

equipped with photovoltaic panels and repair kits for bicycles, while no gold and silver stations offer this service. Railway stations are easily accessible for bikes, reaching values above 90% in all three categories. Toilets inside the station are present only in a bronze station, while they are outside the structure in all the other stations.

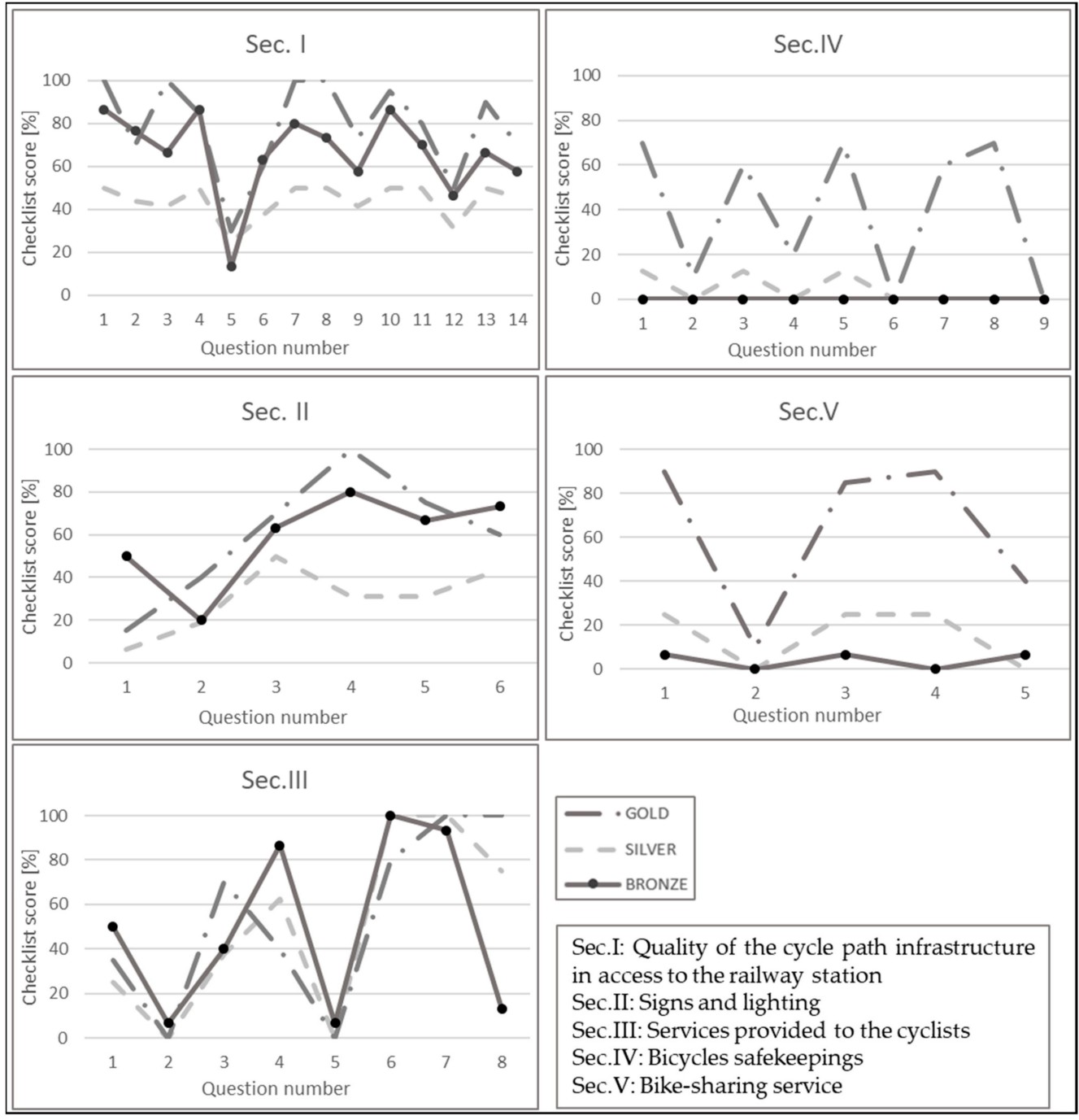

**Figure 4.** Average results for every question of each section of the checklist (see the number of questions in Appendix A).

Sec.IV, relative to the presence of bicycle storage or velostations, obtains a score equal to 0 in all bronze stations since they are absent in all the fifteen stations analyzed. At silver stations, bike storage services are only outside the station, over 50 m away. Gold stations have more deposits, even more than one per station, but all for a fee.

Bike-sharing services, Sec.V, are present in most gold stations but only one of these also offers other micromobility services. At silver and bronze stations, these services are definitely absent. However, the service does not meet the needs of cyclists at all stations and the bicycle must be brought back to a specific hub without the possibility of leaving it where it is most convenient for users.

## 5. Discussion and Conclusions

The main objective of this study is to enhance bike–train intermodality in order to improve air quality and reduce traffic pollution sources. To achieve this goal, the characteristics of the stations are analyzed and classified in gold, silver and bronze according to the RFI classification of the Emilia-Romagna region adhering to PAIR 2020, thus highlighting strengths and weaknesses. The monitoring checklist provided in this study allows for an accurate assessment of the strengths and weaknesses of each mobility hub, identifying aspects to be improved with respect to the city that, following the analysis, presents the most efficient infrastructure.

Gold stations have the best overall quality compared to silver and bronze stations. In fact, they obtain the highest scores in all five categories. Perhaps this is because gold railway stations have the highest number of passengers and must ensure a high standard of services accordingly. In addition, gold stations have the important feature of being located in strategic areas relative to city centers and are part of an intermodal system for travelers living in the area. Considering that in the studied region, commuters are over half of the total population (55.4%) (ISTAT commuter data), it is clear why the most powerful railway stations are located in the most populated areas. Eight out of ten railway stations classified as gold have more than 100,000 inhabitants. In total, the average population of gold stations is approximately 145,000 inhabitants, while the average population of silver stations is approximately 41,000 inhabitants and the average population of bronze stations is approximately 23,000 inhabitants. Silver stations have a lower overall quality than bronze ones, although their population is higher. Considering their proximity to important industrial areas, they should guarantee a better standard of service. However, it should be pointed out that in silver stations, the average population is only 18,000 inhabitants higher than in bronze stations, while they have 104,000 fewer citizens than in gold railway stations.

Analyzing the gap between the two stations, silver railway stations do not reach the rates of the bronze stations in "Sec.I" and "Sec.II". This is because half of the railway stations classified as silver have no infrastructure to facilitate the cyclist to reach them. In the four silver railway stations considered, there are no cycle paths, signs, lighting or speed limits of 30 km/h or less that facilitate access to the railway stations for cyclists. As for the bronze railway stations, only two out of fifteen railway stations have no accessible infrastructure outside the railway stations.

"Sec.V" shows that in bronze stations, bike-sharing services are absent in almost all stations. These stations are located in rural areas where citizens use more private vehicles to move because of the distance from urban centers. A study carried out in rural municipalities in Germany has shown that users living in these areas are interested in shared mobility solutions, but it is up to the local authorities to motivate them with educational work by advertising the service offered and how to use it [51].

Providing bike-sharing services close to railway stations is one of the possible solutions to encourage the use of sustainable means of transport for intermodal travel. Jamber et al. have shown that in areas with numerous public transport stops, there is also a significantly higher number of bike-sharing trips. In addition, the combined use of bike-sharing and train seems to be preferred over bike-sharing and bus [52].

To encourage the use of bike-sharing, adequate infrastructure is needed to reach the railway stations. A study carried out in Maryland has shown that the presence of a well-structured network of cycle paths connected to the main nodes would facilitate the use of bicycles to reach very busy places [53]. It has also been shown that increasing bike racks and other services for users, such as lockers inside or near the railway station, increases the number of cyclists using intermodality between bicycle and train as a means of transport [54]. Additionally, Cervero et al. have shown that the improvement of facilities dedicated to cyclists in or near a railway station is directly linked to the increase in the number of bicycles arriving at the railway station [55]. Bicycle parking should be implemented as well. It has been studied that limited bicycle parking racks near railway stations reduce the number of bicycle users [56].

Finally, gold stations, with a total rate of 67%, should be improved, starting from signs and lighting outside and inside the railway station; for example, by redoing the boundaries of the cycle path, painting it and scheduling regular maintenance. In fact, poor lighting results in a negative perception by users of safety, security and comfort [57]. Moreover, the presence of safe and free parking spaces where cyclists can leave their bikes and sharing services (bikes and e-scooters) outside the station increases comfort and services for cyclists [58].

If two railway stations obtain the same score, the subsections of the checklist will be compared quantitatively according to the number of services. This will result in the railway station with the lowest score in the subsections, namely, the one that needs to be given priority regarding implementing interventions. This method helps to define the most urgent interventions among different railway stations. However, it does not help to establish a threshold of complete satisfaction by users of bicycles as a means of transport, while a 100% threshold would be ideal for each station. To overcome this challenge, future developments of the research foresee the involvement of the main stakeholders through questionnaires and round tables to know their needs and the major criticalities they find daily in the railway stations analyzed. In addition to technical aspects, during the decision-making process, social, economic and environmental aspects should also be considered to define the priority actions for the proposal of the project [59,60]. Moreover, future studies will also include the analysis and comparison of the data on bike–train users with the results of the checklists.

**Author Contributions:** Formal analysis, investigation, data curation, M.P. and C.L.; methodology, M.P. and C.L.; writing—original draft preparation, M.P. and A.Z.; writing—review and editing, M.P., A.Z., C.L., A.S. and H.I.; visualization, M.P. and C.L.; supervision, M.P., A.Z., C.L., A.S. and H.I. All authors have read and agreed to the published version of the manuscript.

**Funding:** This research received no external funding.

**Institutional Review Board Statement:** Not applicable.

**Informed Consent Statement:** Not applicable.

**Data Availability Statement:** Not applicable.

**Acknowledgments:** The authors would like to thank the Emilia-Romagna region and, in particular, Eng. Katia Raffaelli and Lucia Ramponi, for their support and for providing such an interesting and complex case study to be analyzed within the PREPAIR—LIFE15 IPE IT013 project.

**Conflicts of Interest:** The authors declare no conflict of interest.

## Appendix A. The Checklist Used to Evaluate Bike–Train Intermodality and Accessibility

**Table A1.** Checklist of bike-train intermodality and accessibility.

| Section | | Characteristic | | Rate |
|---|---|---|---|---|
| QUALITY OF THE CYCLE PATH INFRASTRUCTURE IN ACCESS TO THE RAILWAY STATION (Sec.I) | 1 | The cycle track close to the MH is | present or the speed limit is 30 km/h | 2 |
| | | | planned | 1 |
| | | | missing | 0 |
| | 2 | The cycle track is located | close to the station | 2 |
| | | | at least 50 m by the railway station | 1 |
| | | | more than 50 m from the railway station | 0 |
| | 3 | The main pavement is | asphalt or similar | 3 |
| | | | self-locking bricks or similar | 2 |
| | | | macadam or similar | 1 |
| | | | natural soil | 0 |
| | 4 | The quality of pavement is | excellent | 2 |
| | | | good | 1 |
| | | | bad | 0 |
| | 5 | The type of cycle track, if present, is | one way or 30 km/h each way | 1 |
| | | | two-ways | 0 |
| | 6 | The cycle track is | reserved to bicycles or adjoining to sidewalk | 2 |
| | | | combined with pedestrians or with 30 km/h | 1 |
| | | | not defined | 0 |
| | 7 | Are geometrical standards respected when combined with sidewalk? | yes | 1 |
| | | | no | 0 |
| | 8 | Is the slope of the of the cycle track below 5%? | yes | 1 |
| | | | no | 0 |
| | 9 | The sides of the cycleway have | a 20 cm edge beam or dedicated lane | 3 |
| | | | white and yellow stripes or more in general a yellow preferential lane | 2 |
| | | | white lame/30 km/h speed limit lane | 1 |
| | | | missing | 0 |
| | 10 | One-off obstacles, reduction in standards are | missing | 2 |
| | | | almost missing | 1 |
| | | | frequent | 0 |
| | 11 | Lateral conflict due to the presence of shops, household entrances or vehicle entrances are | missing | 2 |
| | | | soft | 1 |
| | | | very frequent | 0 |
| | 12 | Conflict with pedestrian traffic | missing | 2 |
| | | | soft | 1 |
| | | | very frequent | 0 |
| | 13 | Conflict with vehicles traffic | missing | 2 |
| | | | soft | 1 |
| | | | very frequent | 0 |
| | 14 | Accessibility: visible and in good state accesses of cycle path | excellent | 3 |
| | | | good | 2 |
| | | | poor | 1 |
| | | | bad | 0 |
| TOTAL | | | | |

**Table A1.** *Cont.*

| Section | | Characteristic | | Rate |
|---|---|---|---|---|
| SIGNS AND LIGHTING (Sec.II) | 1 | The cycle path is painted (or has colored tiles) characterized with | good color | 2 |
| | | | average color | 1 |
| | | | no color | 0 |
| | 2 | Street signs indicating the correct behavior cyclists must follow | existing/makes sense | 2 |
| | | | existing | 1 |
| | | | missing | 0 |
| | 3 | Horizontal street signage: the limits of the cycle path are | very visible | 2 |
| | | | averagely visible | 1 |
| | | | not sufficient | 0 |
| | 4 | The cycle path is provided with | great lighting | 2 |
| | | | normal lighting | 1 |
| | | | poor lighting | 0 |
| | 5 | The type of lighting is | inside the cycle path | 2 |
| | | | on the public street | 1 |
| | | | private, from advertising | 0 |
| | 6 | Does the vertical signage exist in the node? | yes, visible and close to the cycle path | 2 |
| | | | yes, not clearly visible | 1 |
| | | | no | 0 |
| TOTAL | | | | |
| SERVICES PROVIDED TO THE CYCLIST (Sec.III) | 1 | The bicycles parking slots are protected with overhead shelter | with great coverage, 2 or more lanes of bicycles | 2 |
| | | | average coverage, one lane of bicycles | 1 |
| | | | missing | 0 |
| | 2 | If present, are the shelters covered with fotovoltaic panels? | yes | 1 |
| | | | no | 0 |
| | 3 | In case of no shelters, the bike racks are | both types | 2 |
| | | | high, the tire and body of the bicycle can be secured | 1 |
| | | | low, only the tire can be secured | 0 |
| | 4 | Is the number of bike racks enough compared to the needs? | yes | 1 |
| | | | no | 0 |
| | 5 | Is there a bike repair kit? | yes | 1 |
| | | | no | 0 |
| | 6 | Are there locations for leaving the bike? | yes | 1 |
| | | | no | 0 |
| | 7 | Is there a ramp or a lift to descend? | yes | 1 |
| | | | no, only stairs | 0 |
| | 8 | Are there toilets inside the railway station? | yes | 1 |
| | | | no | 0 |
| TOTAL | | | | |

**Table A1.** *Cont.*

| Section | | Characteristic | | Rate |
|---|---|---|---|---|
| BICYCLES SAFEKEEPINGS (Sec.IV) | 1 | Is there a deposit for bicycles? | yes | 2 |
| | | | in plan | 1 |
| | | | no | 0 |
| | 2 | The safekeeping is | inside a covered building | 1 |
| | | | in the open air | 0 |
| | 3 | If present, is the safekeeping within 50 m from the railway station? | yes | 1 |
| | | | no | 0 |
| | 4 | Is there more than one safekeeping? | yes | 1 |
| | | | no | 0 |
| | 5 | Inside the safekeeping, the racks are disposed in | two lanes with central access | 1 |
| | | | one lane with lateral access | 0 |
| | 6 | The safekeeping is | for free | 1 |
| | | | for payment | 0 |
| | 7 | The safekeeping | has services for users | 1 |
| | | | does not have any service | 0 |
| | 8 | The safekeeping | has services for bike repair/bike rent | 1 |
| | | | does not have services for bike repair/bike rent | 0 |
| | 9 | Is the number of available spots shown from the outside? | yes | 1 |
| | | | no | 0 |
| TOTAL | | | | |
| BIKE SHARING SERVICE (Sec.V) | 1 | Is there bike sharing? | yes | 1 |
| | | | no | 0 |
| | 2 | Are there other micromobility services? | yes | 1 |
| | | | no | 0 |
| | 3 | Bike-sharing service is located | less than 50 m from the railway station | 2 |
| | | | between 50–500 m from railway station | 1 |
| | | | more than 500 m from railway station | 0 |
| | 4 | Is the bike-sharing service adequate compared to the needs? | yes | 1 |
| | | | no | 0 |
| | 5 | Can the shared bike be left in a place other than the collection place? | yes | 1 |
| | | | no | 0 |
| TOTAL | | | | |

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
