# Peer review of "Evaluation of Railway Station Infrastructure to Facilitate Bike–Train Intermodality"

_sustainability, doi:10.3390/su15043525_

Round 1
Reviewer 1 Report
This is a comprehensve (detailed case study), informative Technical Paper that highlights the overall contribution of cycling to sustainability, and the importance of overall intermodality. No hesitation in recommending publication. The Reviewer lives in the Greater Toronto Area (GTA) where significant applied studies have been completed for Cycling Handbooks that also include intermodality. It is great to see all of the cycling. The Reviewer enjoyed reading the Paper and will be using it as a key Reference too with Teaching.
Author Response
This is a comprehensive (detailed case study), informative Technical Paper that highlights the overall contribution of cycling to sustainability, and the importance of overall intermodality. No hesitation in recommending publication. The Reviewer lives in the Greater Toronto Area (GTA) where significant applied studies have been completed for Cycling Handbooks that also include intermodality. It is great to see all of the cycling. The Reviewer enjoyed reading the Paper and will be using it as a key Reference too with Teaching.
We would like to thank the Reviewer for the kind comment on our contribution ‘sustainability-2201416’ submitted for the Journal of Sustainability – Section: Sustainable Transportation – Special Issue: Sustainable Transport Solutions and Construction Materials for Safer and Green Roads.
With best regards on behalf of all authors,
Margherita Pazzini

Reviewer 2 Report
The authors present the methodology and the results of the classification and analysis of 33 railway stations in the Emilia-Romagna region (Italy), performed within the PREPAIR project. They have defined a technical checklist to quantify the quality i.e. the criticalities of the infrastructure and accessibility of railway stations to the use of bicycles.
In Ch1, the authors give a short overview of how reducing the use of private cars by the development of an intermodal transport network and mobility hubs could help reduce CO2 emissions, emphasize the benefits of bicycling, and suggest the importance of bike-train intermodality. They continue with the extensive and current state of the research on the quality of the cycling infrastructure and its influence on the propensity to cycle. They highlighted the objectives of “PREPAIR” project and presented research.
In Ch2, the authors present the basic characteristics of the analysed area (The Emilia-Romagna Region) regarding its location, spatial features, trip generators, and classification of its 33 railway stations into three categories considering daily attendance, level of passenger service, areas open to the public, and intermodality.
In Ch3, the authors present the composed checklist of 42 features regarding bike-train intermodality and accessibility of the facilities at the railway stations, divided into 5 main categories (sections), the principles on which they have rated them, and the method for their quantitative evaluation e.i. the overall station's score (quality percentage) calculation.
In Ch3, the authors present the results of the assessment of the strengths and weaknesses of each hub regarding bike-train intermodality and accessibility.
In Ch4, the authors identify the aspects that should be improved on analysed locations.
General concept comments
The manuscript is clear and presented in a well-structured manner, the cited references are current, the data is interpreted appropriately and consistently throughout the manuscript, and the manuscript’s results are reproducible based on the details given in the methods section.
The checklist is effective in rating the railway stations regarding bike-train intermodality and accessibility, it is practical and could be easily used in similar cases. However, the scientific value of this paper is questionable, as it reads as a mere professional report on infrastructure characteristics of specific 33 train stations in Italy. The authors declare that the objective of their article is to define a methodology to give priority to infrastructure needs and interventions to facilitate the accessibility of bicycles in railway stations, but give no hypothesis to be tested. This problem is most obvious from the discussion section of the paper, where authors should have discussed how their investigation results can be interpreted from the perspective of previous studies. Here, in lines 460-480 the authors have attempted to do so and failed since the referenced research elaborates on the influence of infrastructure quality on traffic.
The investigation presented in this paper is therefore incomplete. Authors should at least correlate their classification results with actual numbers of intermodal bike-train users of the stations in question, to verify their method. It would be even better if they could validate the method further by performing it with their project partners on Slovenian stations.
Specific comments
Multiple missing punctuation and spaces between characters.
Consecutive referencing in separate brackets, not according to instructions in the template.
Double sentencing (lines 62-66).
Author Response
REVIEWER#2
We would like to thank the Reviewer for the comments and suggestions on our contribution ‘sustainability-2201416’ submitted for the Journal of Sustainability – Section: Sustainable Transportation – Special Issue: Sustainable Transport Solutions and Construction Materials for Safer and Green Roads. We have carefully taken the reviewer’s comments into consideration and addressed them all in the revised manuscript. Hereafter we provide a point-by-point response.
With best regards on behalf of all authors,
Margherita Pazzini
The authors present the methodology and the results of the classification and analysis of 33 railway stations in the Emilia-Romagna region (Italy), performed within the PREPAIR project. They have defined a technical checklist to quantify the quality i.e. the criticalities of the infrastructure and accessibility of railway stations to the use of bicycles.
In Ch1, the authors give a short overview of how reducing the use of private cars by the development of an intermodal transport network and mobility hubs could help reduce CO2 emissions, emphasize the benefits of bicycling, and suggest the importance of bike-train intermodality. They continue with the extensive and current state of the research on the quality of the cycling infrastructure and its influence on the propensity to cycle. They highlighted the objectives of “PREPAIR” project and presented research.
In Ch2, the authors present the basic characteristics of the analysed area (The Emilia-Romagna Region) regarding its location, spatial features, trip generators, and classification of its 33 railway stations into three categories considering daily attendance, level of passenger service, areas open to the public, and intermodality.
In Ch3, the authors present the composed checklist of 42 features regarding bike-train intermodality and accessibility of the facilities at the railway stations, divided into 5 main categories (sections), the principles on which they have rated them, and the method for their quantitative evaluation e.i. the overall station's score (quality percentage) calculation.
In Ch3, the authors present the results of the assessment of the strengths and weaknesses of each hub regarding bike-train intermodality and accessibility.
In Ch4, the authors identify the aspects that should be improved on analysed locations.
General concept comments
The manuscript is clear and presented in a well-structured manner, the cited references are current, the data is interpreted appropriately and consistently throughout the manuscript, and the manuscript’s results are reproducible based on the details given in the methods section.
- The checklist is effective in rating the railway stations regarding bike-train intermodality and accessibility, it is practical and could be easily used in similar cases. However, the scientific value of this paper is questionable, as it reads as a mere professional report on infrastructure characteristics of specific 33 train stations in Italy. The authors declare that the objective of their article is to define a methodology to give priority to infrastructure needs and interventions to facilitate the accessibility of bicycles in railway stations, but give no hypothesis to be tested. This problem is most obvious from the discussion section of the paper, where authors should have discussed how their investigation results can be interpreted from the perspective of previous studies. Here, in lines 460-480 the authors have attempted to do so and failed since the referenced research elaborates on the influence of infrastructure quality on traffic.
The investigation presented in this paper is therefore incomplete. Authors should at least correlate their classification results with actual numbers of intermodal bike-train users of the stations in question, to verify their method. It would be even better if they could validate the method further by performing it with their project partners on Slovenian stations.
The authors thank the reviewer for the suggestion. In paragraph 5 "Discussion and Conclusions", line 463-484, the authors have modified the references inserted comparing the results obtained with previous studies concerning bike-train accessibility in railway stations.
Data on the number of intermodal bike-train users of the stations in question are not available yet.
The authors would like to point out that, regardless of the flow data, the study focuses mainly on the implementation of infrastructure to promote bike-train intermodality. Future studies will certainly include the analysis of the data obtained with the checklists made.
Specific comments
The authors thank the reviewer for noting the inaccuracies and underlining them. All the editorial issues below (points 2-4) have been corrected.
- Multiple missing punctuation and spaces between characters.
- Consecutive referencing in separate brackets, not according to instructions in the template.
- Double sentencing (lines 62-66).

Reviewer 3 Report
The submitted paper is interesting, dealing with the topic of the possibilities of a bicycle commuter network to railroad stations in one of the regions of Italy. The structure of the work is correct, adequately selected in terms of quantity, quality and subject matter is also the literature, but it should be supplemented with several works. The language of the work is very good, although there are isolated faults. Notwithstanding the above, I request changes, corrections and additions listed below before publication.
General remarks
1. In multiple places of the paper gold, silver are either with small or capital letter. Please standarize
2. References are in different styles and should be standarized according to Journal’s style. In reference list names of all authors should be listed, in several papers they’re referred to as ‘et al.’.
3. It is advised to further discuss the criteria taken to assess the station quality. For instance, it seems reasonable to discuss specifical mobility in rural areas as presented in paper https://www.mdpi.com/2624-6511/5/4/62 or connection between trasport stops and bike sharing https://www.mdpi.com/2673-7590/2/3/38
Detailed remarks
Line 26 and following: lower case 2 in CO2
Line 32: 2050_space_[1]. Error repeated in 34, 36, 49 and many others
Line 46: [13-15] has grey background, few other lines as well
Line 62-65: sentence repeated
Line 74: not only the distance to central business district, but also to other important traffic generators, including high density housing areas
Line 80: consider
Line 82: first use of AHP abbreviation; please explain
Line 92: Osama and Sayed [39]
Line 125: please check Journal’s policy on full stop after title
Line 158: The
Line 164, 196: word[space]-[space]word
Line 208: it is advised to explain criteria of choosing the ‘most performing ideal’
Line 210-211: here and in whole paper it’s unclear if the speed should be reduced to ‘30 and less’ or ‘below 30’ which is not the same
Line 241-252: issue of bike lanes and paths quality should be discussed wider that basing on Regional Guidelines only. Their safety depends also on gradient, lane width, radii of curves etc. In many countries unidirectional bike traffic isn’t an important condition. A good source to support the discussion could be the FHWA bike guide: https://www.fhwa.dot.gov/environment/bicycle_pedestrian/publications/separated_bikelane_pdg/separatedbikelane_pdg.pdf
Line 252: this is not only to limit the speed, but also to enable riding uphill for less agile users
Line 273, 312: double punctor
Line 365: text says 50%, graph says 40
Line 368: ‘station’ instead ‘section’
Line 369: safe keeping of the bike looses importance with increased amount of bikesharing
Line 380: in order to keep a proper performance of a bikesharing system, certain amount of users is required. This number of users is hard to obtain in smaller towns and villages. It’s arguable whether bikesharing is a universal solution and should be promoted in all localties
Line 449-450: isn’t it a psychological factor, because the expectations for bronze stations are simply lower?
Line 460-462: not all members of groups mentioned here fit the previously assumed work / education model
Line 466: beginning of the sentence is repeated
Line 473: ‘road seat’ unclear
Author Response
We would like to thank the Reviewer for the comments and suggestions on our contribution ‘sustainability-2201416’ submitted for the Journal of Sustainability – Section: Sustainable Transportation – Special Issue: Sustainable Transport Solutions and Construction Materials for Safer and Green Roads. We have carefully taken the reviewer’s comments into consideration and addressed them all in the revised manuscript. Hereafter we provide a point-by-point response.
With best regards on behalf of all authors,
Margherita Pazzini
The submitted paper is interesting, dealing with the topic of the possibilities of a bicycle commuter network to railroad stations in one of the regions of Italy. The structure of the work is correct, adequately selected in terms of quantity, quality and subject matter is also the literature, but it should be supplemented with several works. The language of the work is very good, although there are isolated faults. Notwithstanding the above, I request changes, corrections and additions listed below before publication.
General remarks
- In multiple places of the paper gold, silver are either with small or capital letter. Please standardize
The author thank you for having pointed it out. All the stations’ names have been standardized with small letters.
- References are in different styles and should be standarized according to Journal’s style. In reference list names of all authors should be listed, in several papers they’re referred to as ‘et al.
The authors thank you for having noted and pointed it out. The references have been standardized according to Journal’s style.
- It is advised to further discuss the criteria taken to assess the station quality. For instance, it seems reasonable to discuss specifical mobility in rural areas as presented in paper https://www.mdpi.com/2624-6511/5/4/62 or connection between trasport stops and bike sharing https://www.mdpi.com/2673-7590/2/3/38
The author thanks the reviewer for the very interesting papers suggested for the discussion. The references have been added in lines 468-473.
Detailed remarks
The authors thank you for noting the inaccuracies and underlining them. All the editorial issues (points 1-19) below have been corrected.
- Line 26 and following: lower case 2 in CO2
- Line 32: 2050_space_[1]. Error repeated in 34, 36, 49 and many others
- Line 46: [13-15] has grey background, few other lines as well
- Line 62-65: sentence repeated
- Line 74: not only the distance to central business district, but also to other important traffic generators, including high density housing areas
- Line 80: consider
- Line 82: first use of AHP abbreviation; please explain
- Line 92: Osama and Sayed [39]
- Line 125: please check Journal’s policy on full stop after title
- Line 158: The
- Line 164, 196: word[space]-[space]word
- Line 208: it is advised to explain criteria of choosing the ‘most performing ideal’
- Line 210-211: here and in whole paper it’s unclear if the speed should be reduced to ‘30 and less’ or ‘below 30’ which is not the same.
- Line 252: this is not only to limit the speed, but also to enable riding uphill for less agile users
- Line 273, 312: double punctor
- Line 365: text says 50%, graph says 40
- Line 368: ‘station’ instead ‘section’
- Line 369: safe keeping of the bike looses importance with increased amount of bikesharing
- Line 466: beginning of the sentence is repeated
- Line 241-252: issue of bike lanes and paths quality should be discussed wider that basing on Regional Guidelines only. Their safety depends also on gradient, lane width, radii of curves etc. In many countries unidirectional bike traffic isn’t an important condition. A good source to support the discussion could be the FHWA bike guide:https://www.fhwa.dot.gov/environment/bicycle_pedestrian/publications/separated_bikelane_pdg/separatedbikelane_pdg.pdf
The authors thank the reviewer for the very interesting document suggested. The reference has been added in line 247.
- Line 380: in order to keep a proper performance of a bikesharing system, certain amount of users is required. This number of users is hard to obtain in smaller towns and villages. It’s arguable whether bikesharing is a universal solution and should be promoted in all localties
In order to make the study homogeneous and to compare the three categories of stations, gold, silver, and bronze, all sections of the checklist were considered in the analysis. The bike-sharing services were also considered for bronze stations despite the number of users being definitely lower than the silver and gold stations. The reviewer’s suggestion will certainly be the starting point for future studies where the stations considered will be analyzed in more depth.
- Line 449-450: isn’t it a psychological factor, because the expectations for bronze stations are simply lower?
Many silver stations are located near important industrial areas and, although the population is only 18,000 more than bronze stations, their quality is lower. Providing services for users departing from or reaching neighboring logistics hubs is a fundamental aspect in order to discourage users from using private vehicles to make journeys. This aspect has been clarified in the paper in lines 452-453.
- Line 460-462: not all members of groups mentioned here fit the previously assumed work/education model.
The authors thank the reviewer for the advice. The reference has been modified.
- Line 473: ‘road seat’ unclear
The authors modified this part of paragraph 5 “Discussion and Conclusions” by inserting new references on bike-train intermodality and accessibility to make the discussion clearer (lines 463-484).

Reviewer 4 Report
The research is very interesting, the methodological approach is straightforward and well outlined. There are few typos in the manuscript and in certain sentences the English is not proper. (for example lines 29-30).
Overall, the text is well done and apart from these minor adjustments, does not need any further editing before publication.
A small clarification, is it right to define a "commuter" someone who is moving within its own city (line33)?
Author Response
We would like to thank the Reviewer for the comments and suggestions on our contribution ‘sustainability-2201416’ submitted for the Journal of Sustainability – Section: Sustainable Transportation – Special Issue: Sustainable Transport Solutions and Construction Materials for Safer and Green Roads. We have carefully taken the reviewer’s comments into consideration and addressed them all in the revised manuscript. Hereafter we provide a point-by-point response.
With best regards on behalf of all authors,
Margherita Pazzini
- The research is very interesting, the methodological approach is straightforward and well outlined. There are few typos in the manuscript and in certain sentences the English is not proper. (for example lines 29-30). Overall, the text is well done and apart from these minor adjustments, does not need any further editing before publication.
The authors thank you for the suggestion. All manuscript has gone through new editing by a technical English native Agency.
- A small clarification, is it right to define a "commuter" someone who is moving within its own city (line33)?
The authors thank the reviewer for having pointed it out. They substituted “commuters” with “travellers”.

Round 2
Reviewer 2 Report
The authors have successfully modified the discussion and compared their results with previous studies.
As the number of intermodal bike-train users of the stations is not yet available, authors should add in their discussion that future studies should include the analysis and comparison of the data on bike-train users with the results of their checklists.
I suggest modifying the title to Evaluation of railway station infrastructure to facilitate bike-train intermodality: A case study of the Emilia-Romagna region.
Author Response
#REVIEWER 2 – second round
The authors have successfully modified the discussion and compared their results with previous studies.
As the number of intermodal bike-train users of the stations is not yet available, authors should add in their discussion that future studies should include the analysis and comparison of the data on bike-train users with the results of their checklists.
The authors thank the reviewer for the advice. At the end of paragraph 5 “Discussion and Conclusions” the authors have added a note on future studies (lines 503-504).
I suggest modifying the title to Evaluation of railway station infrastructure to facilitate bike-train intermodality: A case study of the Emilia-Romagna region.
The authors thank the reviewer for the useful suggestion. In order to have a large spread of the research the authors have evaluated that it is better not to specify the case study. The aim is to propose a replicable methodology that can be applied in similar case studies. The Emilia-Romagna region is only an example of an application to better understand the analysis approach.
